# INGAP-Peptide Variants as a Novel Therapy for Type 1 Diabetes: Effect on Human Islet Insulin Secretion and Gene Expression

**DOI:** 10.3390/pharmaceutics14091833

**Published:** 2022-08-31

**Authors:** James M. Porter, Léa Guerassimoff, Francisco Rafael Castiello, André Charette, Maryam Tabrizian

**Affiliations:** 1Department of Biological and Biomedical Engineering, Faculty of Medicine and Health Sciences, McGill University, Montreal, QC H3A 0G4, Canada; 2Campus MIL, l’Université de Montréal, Montreal, QC H2V 0B3, Canada; 3Faculty of Dental Medicine and Oral Health Sciences, McGill University, Montreal, QC H3A 1G1, Canada

**Keywords:** islet transplantation, peptide therapeutics, insulin secretion, gene expression

## Abstract

Islet transplantation offers a long-term cure for Type 1 Diabetes (T1D), freeing patients from daily insulin injections. Therapeutic peptides have shown potential to increase the insulin output of pancreatic islets, maximizing the impact of grafted cells. The islet neogenesis-associated protein (INGAP), and its bioactive core (INGAP-P), stimulate beta-cell function and viability, offering the possibility for islet treatment prior to implant. However, dosing efficacy is limited by low circulation time and enzyme degradation. This proof-of-concept study presents the investigation of novel molecular variants of INGAP-P to find a more bioactive form. Custom-designed peptide variants of INGAP-P were synthesized and tested for their effect on the insulin secretion and gene expression of live human islets. We exposed the live islets of five donors to varying glucose concentrations with INGAP-P variants in solution. We identified four peptide variants (I9, I15Tyr, I19 and I15Cys) which displayed statistically significant enhancements over negative controls (representing a 1.6–2.8-fold increase in stimulation index). This is the first study that has assessed these INGAP-P variants in human islets. It highlights the potential for customized peptides for type 1 diabetes therapy and provides a foundation for future peptide-screening experiments.

## 1. Introduction

Three major umbrellas of diabetes include autoimmune mediated beta-cell-specific destruction—type 1 diabetes mellitus (T1D) mellitus, metabolic exhaustion of pancreatic endocrine cells due to acquired insulin resistance—type 2 diabetes (T2D) mellitus, and the breakdown of insulin-producing cells by inflammatory effects of chronic exocrine pancreatic disease in type 3c diabetes (T3cD) mellitus [1,2,3]. Together, these forms of diabetes affect nearly half a billion people [4]. These various pathologies of the diabetic state reflect the major needs to be addressed in terms of cell replacement therapies. Islet grafts must be protected from host-immune rejection, not only to ensure insulin independence but to protect from the adverse effects of long-term immunosuppression [5]. Islets must have stable vasculature, usually provided by integration with the basement membrane via endothelial cells and pericytes [6,7]. Adequate perfusion of blood and oxygen is essential for islets’ metabolism, efficient hormone delivery and paracrine communication [7]. Finally, inflammatory effects are responsible for the death of 30–50% of implanted islets within the first hours to days of the procedure [8]. Addressing these main aspects of diabetes pathologies is essential to a robust islet transplantation.

For T1D, as with other diseases, both exogenous and endogenous cell replacement therapies are available. Endocrine function can be restored by infusing islets from an outside donor (allograft), by replacing a patient’s own islets following a pancreatomy due to cancer or other injury/inflammatory disease (autograft), or by regenerating the native pancreas. Regeneration can occur through replicating remaining beta cells, generation of new endocrine cells through differentiation—pluripotent pancreatic progenitor cells can be attained through dedifferentiation of acinar ductal cells, and subsequently differentiated into insulin-producing beta cells—or through α to β-cell transdifferentiation [9,10,11,12]. Indeed, the islet neogenesis-associated protein (INGAP), is a member of the RegIII family of hormones produced by the exocrine pancreas that communicate with endocrine cells via acinar/islet paracrine interactions to improve islet survival and function [2].

Insulin is one of ~80 peptide drugs currently on the market, which are known for their selectivity and low toxicity, but are limited by degradation effects and poor stability [13]. Insulin analogues have been engineered for fast- or long-acting release, which together sold $25 billion US in 2019. Injecting the incretin hormone Glucagon-like peptide-1 (GLP1) stimulates insulin production in type 2 diabetics; however, it is rapidly cleared by the kidneys in under 2 min. Exendin-4 (EX4) peptide, extracted from Gila monster venom, agonizes the GLP1 receptor while maintaining circulation over 5 h, though it leads to inflammation and pyknosis of pancreatic acinar cells in rats [14]. Further examples of synthetic peptide drugs can be found in liraglutide and dulaglutide (also GLP1 receptor agonists).

Endocrine targets comprise ~17% of the current peptide therapeutic market, and INGAP is involved with the protection and neogenesis of the native endocrine pancreas by increasing beta-cell mass and insulin secretion [13,15,16]. INGAP could lead to a peptide therapy by emulating the exocrine pancreas to protect and promote islet function, whether a native population or newly implanted cells. Along with prevascularization and immune protection of grafts, peptide therapies are among the current strategies for in vitro enhancement of islets prior to implant. The bioactive subsequence of INGAP was initially identified through modelling to be residues 104–118 [17]. The so-called INGAP-pentadecapeptide (INGAP-P), and particularly its cyclized analogues have been shown to increase β-cell proliferation in RINm5F cells [18]. The mechanism of INGAP-P’s interaction with the beta cell is not completely understood. It is suspected that the beta cell surface receptor of the KiR6.2/SUR1 channels may be a ligand candidate [19]. INGAP-P has been proven safe for humans, already having reached Phase 2 clinical trials, increasing the post-transplant viability of islet allografts, but the short half-life motivates improvement of the drug [15,20,21]. INGAP-P appears to activate the Ras/Raf/ERK pathway, suggesting alteration of the mRNA translation [21]. Since INGAP-P has been shown to increase islet proliferation after 24 h, we focused on the insulin stimulatory and gene regulation properties of the peptide, which has shown to be internalized within 30 min by rat islet RIN-m5F cells, increasing insulin secretion and affecting gene expression in neonatal rat islets [22].

This study investigates the relation between molecular conformation and insulin-promoting capability of INGAP-P on the insulin secretion of human islets. Our aim was to discover which peptide characteristics are most important for maximizing the outcomes of transplanted islets in a clinical setting. We synthesized 7 variations of the 15 amino acid INGAP-P sequence [23] and tested their ability to stimulate insulin secretion in human islets [24]. Live islets from human organ donors were subjected to basal and stimulatory concentrations of glucose, with controls or peptide variants added to the high glucose solutions. Negative control was high-glucose media with no other additives, with known insulin promoter exendin-4 (EX4) or KCl, commonly known for its ability to simulate glucose-induced insulin release, serving as a positive control [25,26]. Supernatants were quantified for insulin content by Enzyme Linked Immunosorbent Assay (ELISA). Islets were subsequently lysed to assess gene regulation of nuclear mRNA expression. A panel of islet genes implicated in the insulin secretion pathway (triggering or amplifying) were evaluated using quantitative reverse-transcription polymerase chain reaction (RT-qPCR), relative to the glyceraldehyde-3-phosphate dehydrogenase (GAPDH) housekeeping gene [15]. Gene expression is compared to insulin secretion for each therapeutic peptide candidate across human donors. Although the preliminary sample size was limited by the availability of human donors, this pilot study provides valuable insights on the variation between human donors and sheds light on the challenges of developing synthetic peptides to improve the outcomes of islet transplant for the treatment of T1D that can face the high variability of insulin secretion patten among individuals.

## 2. Materials and Methods

### 2.1. INGAP-P Variant Peptide Synthesis

Synthesis of custom-designed peptides is detailed in our previous report [23]. Briefly, individual amino acid residues are sequentially added to the peptide chain using an Fmoc solid-phase method first described by Garcia et al. [18]. This Fmoc solid-phase protein synthesis was done using the Focus XC peptide synthesizer from AAPPTec (Louisville, KY, USA), with Rink-Amide 4-methylbenzhydrylamine (MBHA) resin and (Benzotriazol-1-yloxy)tripyrrolidinophosphonium hexafluorophosphate/1-Hydroxybenzotriazole (Py-BOP/HOBt) coupling reagents. Piperidine in dimethylformamide (DMF) solvent is used for deprotecting the resin of the Fmoc group, then an acetyl group is used to cap the N-terminus. Dichloromethane (CH_2_Cl_2_) is used to evaporate the solvents before cleaving the peptide from the resin. Reverse Phase High-Performance Liquid Chromatography (RP-HPLC) was employed to purify peptide products, using a C18 stationary phase column. The polar peptide spends more time in the hydrophobic stationary phase compared to faster-moving impurities, allowing separation by fractionation. A gradient of two polar solvents made up of water with 0.4% formic acid (solvent A), and acetonitrile (solvent B) was used to ensure adequate separation. Solvent B was progressively increased from 10–50%. Final purity was confirmed by comparing the expected exact mass to the measured value via liquid chromatography-mass spectrometry (LC-MS) direct injection analysis. Peptides were kept frozen at −80 °C before overnight lyophilization, allowing for stable storage as powders until dilution for biological testing.

### 2.2. Study Subjects

Human pancreatic islets were obtained from listed organ donors through the United Network for Organ Sharing (UNOS), with informed consent from donors prior to their death. Chosen donors were non-diabetic and confirmed COVID-19 negative. These five consenting donors, between ages of 19 and 59, had known glycated hemoglobin, BMI and cause of death. Medical history and donor condition at the time of death emphasize the highly individual nature of each test subject. Experimental protocols were approved by the McGill Institutional Review Board, with project approval until February 2022. All methods were carried out in accordance with relevant guidelines and regulations. Human pancreatic islets were obtained from organ donors through Prodolabs (Aliso Viejo, CA, USA), within 1–2 days post-isolation. Non-diabetic, COVID-19 negative human subjects were selected between the ages of 19–59 years (informed consent was obtained from all subjects). Individual donor information including BMI, Hb1Ac and cause of death are shown in Table 1. The Appendix A details islet culture and handling methods, conducted following the McGill Human Islet Transplant Lab.

### 2.3. Human Islet Handling and Stimulation

Islets were isolated by Prodolabs (Aliso Viejo, CA, USA) from digested human pancreases at the time of death and purified from acinar tissue. The islets were shipped in Prodo Islet transport media (PIM(T)) at 4 °C and transferred to culture media at 37 °C upon arrival. Full islet recovery and handling protocols were referenced in the Appendix A. Islets were cultured for a total of 2–3 days (at 37 °C and 5% CO_2_), including prior to shipping, before testing. Islet stimulation was carried out within this period to avoid any loss of functionality over time.

GSIS was tested using an in vitro static-well incubation, following the protocol of the McGill Islet Transplant Laboratory (full details in Appendix A). Groups were tested in triplicate, with ~150 islet equivalents (IEQ) in each. Cell counts were verified using Quant-iT PicoGreen DNA Quantitation kit, normalizing to 10.4 ng of DNA/islet equivalent on average.

Briefly, islets were taken from culture media and resuspended in 2.8 mM glucose Krebs buffer (preparation described in Appendix A) to ensure basal insulin secretion rate after removal from 5.8 mM glucose culture medium. Islets remained in the low-glucose solution for 30 min to stabilize prior to testing. Using 8 μm pore cell culture inserts (VWR), islets were transferred to a fresh well also containing 2.8 mM glucose solution and incubated for 1 h at 37 °C. Next, islets were moved into the stimulation well, with 28 mM glucose Krebs, plus peptide variants or controls. As a negative control we used 28 mM glucose Krebs with no other additives. Positive control was 28 mM glucose with 1 μg/mL EX4, or KCl. Exendin-4 is known to be a slow-acting GLP-1 agonist and insulinotropic factor [25,27,28]. Test groups consisted of 7 different variations upon the INGAP-P 15 amino acid sequence. Islets are then recovered from the inserts and lysed or frozen at −80 °C for later qPCR analysis. Insulin was quantified using Mercodia Human ELISA (Cedarlane), within 24 h of sampling. After initial testing, 5-fold sample dilutions were used, plated in duplicate. Optical absorbance was measured at 450 nm with an i3 SpectraMax plate reader (Molecular Devices, San Jose, CA, USA). Aliquots of the supernatant were sampled at the beginning (t = 0) and end (t = 1 h) of the incubations. These two readings were subtracted to find the insulin released during a 1-h incubation. To calculate the stimulation index, the insulin concentration in the stimulation well was divided by that in the low-glucose buffer.

### 2.4. Live Cell Viability Imaging

Live/dead staining for fluorescence imaging was done using the Biotium Viability/Cytoxicity Assay Kit (for Animal, obtained from VWR). The green live dye consisted of an elasterase substrate, which was cleaved into fluorescent calcein, and only remained if the cell membrane was intact. The red dead dye was Ethidium Homodimer—III (EthD-III), a membrane-impermeable DNA dye, only penetrating cells when the cell wall was compromised. Islets were prepared for imaging first by resuspension in serum-free media (here: PBS). Staining media was 2.5 µL calcein AM (hydrolyzed, pH 8) and 10 µL EthD-III in PBS (5 mL total volume). Islets immersed in staining media were shielded from light and incubated at 37 °C for 30 m before imaging. Images were taken using the LSM 710 Confocal Scanning Microscope (excitation wavelengths: 488 nm for live and 543 nm for dead). Image reconstruction and analysis was performed using the Zen Microscope Software by Zeiss.

### 2.5. RT-qPCR Quantification of Islet Gene Expression

Following glucose-stimulated insulin secretion (GSIS), cells were lysed for gene analysis. Nuclear mRNA was extracted and purified using the PureLink RNA Mini Kit (Invitrogen by Thermo Fisher Scientific, Mississauga, ON, Canada) protocol, verified by NanoDrop quantification. Gel electrophoresis confirmed extracted mRNA quality. Purified RNA samples were kept at −80 °C until RT-qPCR with the Luna OneStep reaction kit. mRNA was reverse transcribed into cDNA, then amplified by thermal cycling. Although glyceraldehyde-3-phosphate dehydrogenase (GAPDH) is commonly used as a reference gene, some variation may occur among different human donors [29]. Therefore, target gene expression was quantified relative to the housekeeping gene GAPDH within each group, and further normalized against the negative control (high-glucose stimulation with no peptides added). Table 2 shows selected islet genes related to the insulin detection, metabolism and secretion pathway of β-cells.

### 2.6. Data Analysis

All islet groups were tested in triplicate (*n* = 3). Insulin secretion data were presented as mean ± SD, and One-Way ANOVA was applied for significance.

Relative gene expression for the 5 chosen genes was normalized relative to the house-keeping gene of that group, then each group was normalized against the negative control. The relative expression factor, R, was calculated in this way, by the Livaak-Schmittgen method: $R=2^{-\Delta \Delta Ct}$, where $\Delta \Delta C_{T}=\Delta C_{T\left(test\right)}-\Delta C_{T\left(calibrator\right)}$, with $\Delta C_{T}$ for each gene calculated in reference to the GAPDH housekeeping gene. The calibrator test group was our negative control, a high glucose (28 mM) stimulation well, with no other peptides added.

## 3. Results and Discussion

### 3.1. INGAP-P Variant Synthesis and Characterization

Table 3 shows modified peptides tested for enhancing insulin secretion, based on variations of the 15 amino acid sequence of INGAP-P. Following synthesis (experimental methods Section 2.1), peptides were purified by HPLC to ≥95%. Peptide identification was done by comparing exact expected mass to charge ratios (ex.: tyrosine-modified INGAP-P in Figure 1A) to those measured by LC-MS in Figure 1B (c.f. also Appendix A for mass spectroscopy and for LC-MS analyses).

Results showed *m*/*z* ratios accurate to within 0.09%. Cyclization by connecting the peptide’s N- and C-terminals was shown to greatly improve stability and extend the circulating half-life [13]. Two cyclic modifications to the INGAP-P sequences were prepared in I15Cys and I19Cys. Using SwissModel’s 3D protein folding simulator, images in Figure 1C were created from the I15Tyr sequence. Simulations revealed the protein sequence as an F1 capsule/anchoring protein. Conserved hydrophobic clusters on the surface of the CAF1A usher C-terminal domain were important for antigen assembly. From the visual representation in Figure 1C, we can see a twisting ribbon-like conformation, similar to the helical structures of GLP1 and exendin-4, combined with a self-intersecting loop shape which may contribute to higher stability.

### 3.2. Islet Culture and Peptide-Stimulated Insulin Secretion

Figure 2 shows confocal fluorescence imaging of intact islets from the youngest donor prior to glucose challenges. Live cell imaging revealed good islet morphology and viability following transport, with very few dead cells throughout each layer of the 20 image Z-stack. Staining and imaging procedures are detailed in experimental methods Section 2.4.

Figure 3 shows the insulin concentration of supernatant sampled from basal and stimulation wells, shown as mean ± SD, quantified by ELISA (*n* = 3). Donors 1–3 were subjected to the complete panel of peptide variants, whereas 4 and 5 focused on smaller subsets. Donors 2 and 4 (age 55 and 26) are presented at scale for comparison to other donors, with inset graphs showing zoomed in data plots. Donor 3, age 37, produced an impressive insulin response across all peptide groups, compared to the other donors. The average concentration of insulin in stimulation wells for this donor was 36.4 ng/mL. On the other hand, islets from donor 4, age 26, were not secreting at an appreciable rate. This donor was exposed to a subset of the groups to focus on candidate peptide variants.

Figure 4 displays the stimulation index, calculated as the ratio of insulin in the high-glucose to that in the low-glucose buffer wells, facilitating a comparison of functionality between groups. One-way ANOVA was applied between each peptide group and the negative control. In addition to the higher concentrations of insulin released, donor 3 also displayed the highest fold-change of insulin in response to stimuli. Here, the INGAP-P (I15) peptide displayed an insulin-boosting effect similar to EX4, compared to negative control. Significant improvements were seen for I15Cys (donor 2), and for I9, I15Tyr and I19 (donor 3). The last panel shows the combined stimulation index for all donors across peptide groups.

High glucose islet stimulation has been shown to result in AMPK inhibition in rats, mice and humans [30]. With recent advances in research, we are still only now learning what human rodent islets have specifically in common, and when animal findings can be translated to human applications [31]. Exposing human islets to 16.7 mM glucose-elevated thrombospondin-1 gene expression while decreasing vascular endothelial growth factor (VEGF) mRNA by 20% [32]. Human insulin ELISA quantification for the 5 donors is shown in Figure 3. Donor 2 showed a statistically significant increase in output to I15Cys, and donor 3 had significant enhancement of insulin for I9, I15Tyr, and I19, compared to controls. Other donors had too much variation to identify beneficial peptides. Noteworthy is that two of the younger donors (3 and 5), at ages 37 and 19, respectively, displayed a heightened insulin production in response to all groups, compared to other donors. It is possible the observed increased insulin production of the third and fifth donors was at least a partial consequence of their age at the time of death. Donor 4 (age 26), however, showed little stimulation response, secreting much less insulin than other donors. With a BMI of 30.3, this donor would be considered obese.

From donor information in Table 1, the cause of death for donors 3 and 5 being head trauma suggests that these islets may have been in better physiological condition at the time of isolation, reflected in lower BMI and higher islet size index. In contrast, the older donors 1 and 2 died from stroke. Although interruption of blood flow leads to a hypoxic environment within the brain, this does not drastically affect islet isolation and purification from acinar tissue as much as the cold ischemia time. Islet purity—one of the most critical factors in transplant—was consistent across all donors, showing oxygen conditions in the pancreas were not directly influenced, as reported by the isolation team.

Preliminary perifusion tests of human islets using known secretagogues displayed very much similar release profiles, despite varying degrees of insulin released (Appendix A). Based on these initial observations, it was decided first to focus on quantifying the amount of insulin released during the primary response as a gauge for evaluation. Future works may utilize surface plasmon resonance biosensing or other such kinetic assays to elucidate the dynamics of peptide stimulated insulin release.

### 3.3. RT-qPCR Analysis of Islet Nuclear mRNA

Following islet lysis and nuclear RNA extraction and purification, PCR plates were prepared for amplification by mixing RNA with islet-specific primer sequences to quantify the relative gene expression of each group, in response to stimuli. Purity and quality of RNA was verified by Nanodrop quantification, yielding a 260/280 ratio of 2.09. Gel electrophoresis showed the presence of 28S and 18S RNA, shown in Appendix A. Forward and reverse primers for qPCR were combined and tested for amplification, shown in Table 2. Primer and target RNA amplifications were validated using 1, 5 and 10-fold dilutions (Appendix A). A linear slope of $C_{q}$ vs. log of the starting amount verifies the PCR reaction.

Due to scarcity of human islets for research coupled with a lack of islet cell lines, there is relatively little known of islet gene regulation in response to glucose stimulation [28]. Figure 5 shows expression levels for the panel of insulin secretion pathway genes for each peptide group, for the 5 donors tested. Quantified gene expression levels were found using the Livaak-Schmittgen method relative first to the GAPDH housekeeping gene and then to high-glucose negative control (no peptides added). The colour bar legend on the right of Figure 5 provides the scale of values, with red representing a downregulation, white being unaffected and blue the most upregulated. Donors 1 and 2 showed substantial downregulation of most genes in response to the conserved INGAP-P sequence (I6), and the specific motif (I9). We can see the peptide group for donor 2 having the highest stimulation index (I15Cys), generally upregulated mRNA expression among all islet genes on the target panel, except glucose transporter 2 (Glut2, “G2”, also known as SLC2A2). The donor who displayed the most upregulation across all groups was 3, also releasing the most insulin in GSIS experiments than the other subjects. Donor 3 showed higher relative pancreatic-duodenal homeobox factor-1 (PDX1) expression across all peptide groups as compared to other donors. Generally unaffected, PDX1 was upregulated in response to the I15Gly peptide variant. In contrast, donors 1 and 2 showed downregulation of PDX1 in response to multiple groups. Looking at the gene expression of 5 donors, it appears that INS and PDX1 were the most stable across peptide groups, whereas Glucagon, SUR1 and Glut2 were more affected. I15Gly for donor 3 appears to be most upregulated among all groups, though it did not produce a noticeable difference in secretion of insulin. Donor 3 showed the most upregulation of islet genes related to the insulin secretion pathway and showed remarkably higher insulin in response to all peptides, compared to other donors.

Figure 6 panel A shows the expression of Glut2 and Glucagon genes quantified relative to GAPDH. Donor 5 (age 19) is excluded from this figure since those islets were used for a dose-response study of INS and PDX1 genes (Figure 6C). Glucagon acts to raise blood glucose by stimulating glycogenolysis and gluconeogenesis, whereas Glut2 acts as a glucose sensor and to uptake glucose into the β-cell, initiating the insulin pathway levels [33,34]. By observing the scatter plots of gene expression in Figure 6A, we see that 3 out of 4 donors did indeed exhibit a reciprocal action of these two genes. Donor 3 (age 37), however, remarkably displayed heightened glucagon expression simultaneous to Glut2. ELISA data shown in Figure 3 and Figure 4 show this same donor secreted the most insulin overall and produced the highest fold change in response to stimuli compared to other donors.

Figure 6B shows the expression of INS and PDX1 genes, averaged across all peptide groups for the 5 donors investigated. An increase in age relates to a corresponding increase in DNA methylation at multiple sites, affecting INS and PDX1 gene regulation [35]. In fact, aging has been connected with increased methylation of human pancreatic islets, resulting in decreased expression of INS and PDX1 genes and greater loss of β-cells with earlier diabetes onset [35,36]. The younger donors exhibited a slightly elevated expression of both INS and PDX1 genes, drastically so in the case of donor 3. Insulin-like growth factor-1 (IGF1) serum concentration reaches islets maximum during puberty and falls off roughly threefold by age 60 [37]. With age differences between donors up to 35 years, this represents a significant factor in the condition and functionality of the donors’ pancreatic islets. As age has been negatively correlated with islet GSIS, lower insulin secretion may result from decreased ability to regenerate β-cell mass [38]. Changes in glucose concentration alone have been shown not to affect the human islet PDX1 gene, therefore observed regulatory changes are expected to result from exposure to INGAP-derivatives, or differences between individual donors. Although donor 4 was also younger in comparison to 1 and 2, the overall insulin response was much lower across all groups. Heterogeneity among islet donors highlights the need for large sample sizes to accurately predict how therapeutic compounds may affect the diabetic population.

Figure 6C shows a mini study of dose-response for 4 peptide variants (I15Tyr, I15Cys, I19 and I19Cys) on INS and PDX1 gene expression. This test was comprised of exposing islet groups to the same peptide at either 1 or 100 µg/mL, as well as negative control. Data shown in the bar graph are the logarithm of fold change in gene expression from 1–100 µg/mL peptide exposure. I15Tyr and I15Cys show similar responses in that increasing peptide concentrations raised PDX1 and lowered insulin expression levels. Both I19 variants showed a steep decrease in PDX1, though they differed in their INS gene responses.

### 3.4. Challenges and Limitations

The primary limitation with identifying therapeutic compounds for T1D therapy, as in this study, is the need for extensive in vitro analysis and in vivo transplantation models. Due to the steps involved with synthesis and initial screening evaluation, along with the number of compounds investigated, the authors found it necessary and reasonable to first identify agents of interest to qualify for in-depth analysis.

Without a clear consensus on the characteristics of rodent cells that can be directly translated to human, we focused on studying primary human islets from organ donors. Difficulties involved with this methodology include the sporadic availability of islets, time of shipping and necessity to place islets into transport media, slowing their metabolism. Once the cells recover, stimulation must be carried out within 1 week to ensure maximal viability and functionality. Most importantly, each individual donor has their own medical history and unique physiology at the time of death, introducing variance from one donor to another. In addition, analysing nuclear lysate for gene expression precludes normalization by cell counting with DNA quantification. Other non-β islet cells would contribute mRNA following lysis, leading to uncertainty of the regulation of insulin secretion related genes. This could be addressed by using immortalized cell lines or separating islet cells to specifically interrogate β-cells. Nonetheless, the overall islet response is a meaningful subject of study in terms of blood glucose regulation.

Consequently, the results presented here are significantly relevant, though initial sample size (*n* = 5) was limited by availability of suitable donors. The standard deviation in this experiment can be used to find how many subjects should be studied to discern the best of 10 groups tested. To lower the required sample group size, we recommend focusing on a subgroup (3 or 4) of the peptides of interest. Based on power calculations using experimental ELISA data, we offer the sample size estimation of *n* = 33. This was found with an online calculator (http://powerandsamplesize.com/Calculators/Compare-k-Means/1-Way-ANOVA-Pairwise-1-Sided, accessed on 1 July 2022), using the equation:(1)$$n_{A}=\left(\sigma_{A}^{2}+\sigma_{\frac{B}{\kappa }}^{2}\right){\left(\frac{z_{1-\frac{\alpha }{\tau }}+z_{1-\beta }}{\mu_{A}-\mu_{B}}\right)}^{2}$$
where:
$\kappa =n_{A}/n_{B}$
is the matching ratio,
σ
is the standard deviation,
$\sigma_{A}$
is the standard deviation of Group “A”,
$\sigma_{B}$
is the standard deviation of Group “B”,
α
is Type I error,
τ
is the number of comparisons to be made, and
β
is Type II error (1−β is the power: 80%). This pilot study can therefore be used to guide future experiment design in choosing peptide candidates and human donor sample sizes.

## 4. Conclusions

Personalized medicine can be made possible through comprehensive diagnostic evaluation accompanied by a bank of knowledge regarding various therapeutic agents and their target effects. Although the preliminary sample size was limited by the availability of human donors, this work demonstrates the ability of INGAP-P to boost insulin secretion, and identifies several variants for further investigation, based on their ability to impact islet signaling genes. In addition, this pilot study provides valuable insights on the variation between human donors and sheds light on the challenges of developing synthetic peptides to improve the outcomes of islet transplant for the treatment of T1D that can face the high variability of insulin secretion patterns among individuals. However, there remains much to learn about the potential benefits for therapeutic peptides and their synthetic variants. In particular, the individual differences between human islet donors coupled with the shortage of native cells necessitate platforms for efficient mapping of regulatory effects of libraries of compounds. Future screening of therapeutic compounds could be conducted on immortalized human insulin producing cell lines (such as EndoC-βH1) in order to reduce variability. The varying reactions each donor displayed to the same media incubations also motivate individually tailored treatment plans. Further testing, aided by high-throughput kinetic-based assays such as surface plasmon resonance imaging would help to characterize peptides such as INGAP-P derivatives for medicinal benefits to reduce the need for insulin injections and improve islet transplant efficacy.

## Figures and Tables

**Figure 1 pharmaceutics-14-01833-f001:**
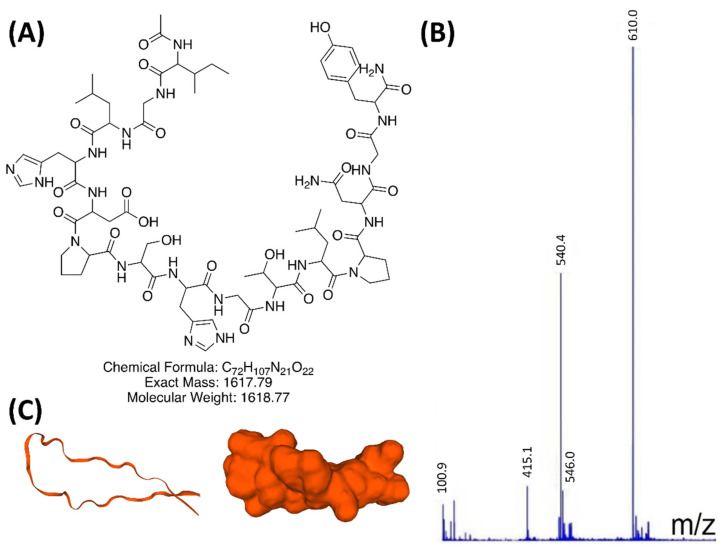
(**A**) I15Tyr, the INGAP-P sequence modified with tyrosine replacing the serine residue at the C-terminal position. (**B**) LC-MS measured mass to charge ratio of 810.0 kg/C, matching the expected value within 0.09%. Table 1 displays the variations made to INGAP-P (“I15”) for testing. (**C**) 3D simulation of conformational folding using the INGAP-P sequence as input. (From SwissModel online tools https://swissmodel.expasy.org/interactive/H396J7/models/, accessed on 1 July 2021).

**Figure 2 pharmaceutics-14-01833-f002:**
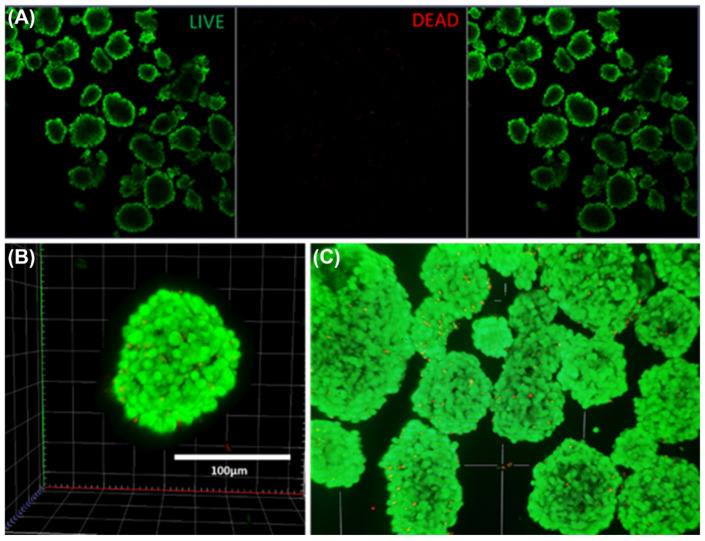
Confocal Live (Green: Calcein) and Dead (Red: EthD-III) fluorescence image of human pancreatic islets. (**A**) Single focal plane Live, Dead and Combined images, (**B**) 3D reconstruction of 20-image Z-stack shows single cell viability throughout a single islet, and (**C**) Consistent viability across a population of islets of various sizes, with small single dead cells on the surface of and detached from living islets.

**Figure 3 pharmaceutics-14-01833-f003:**
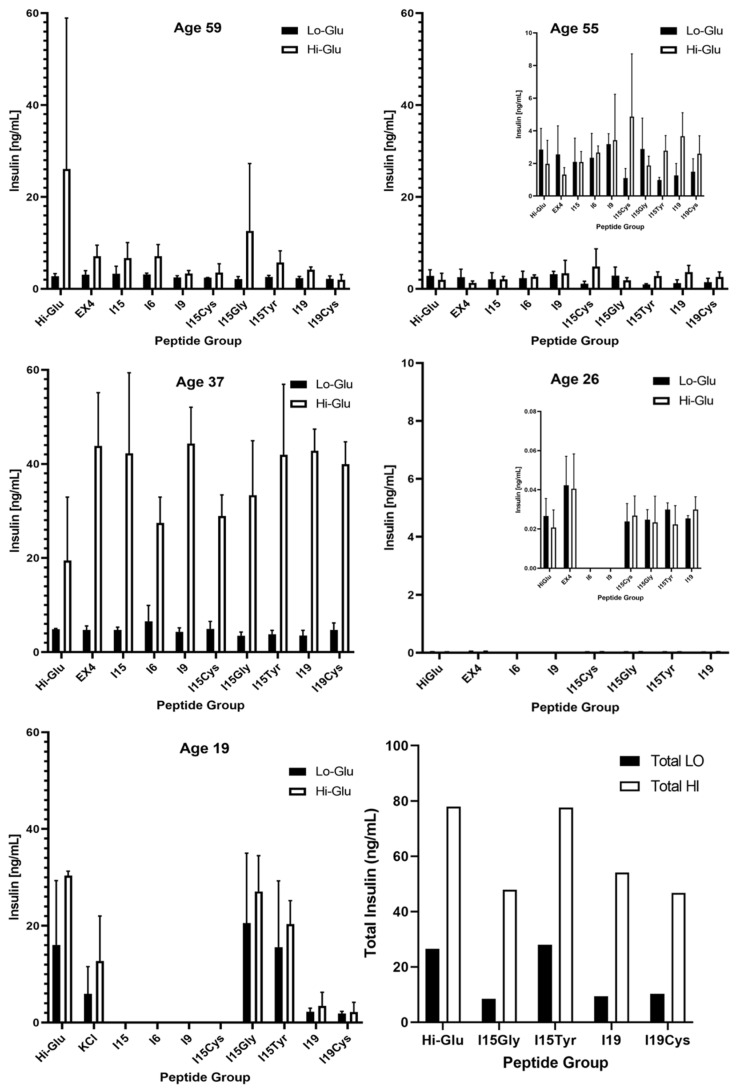
ELISA quantification of islet insulin release in low- and high-glucose wells, for each peptide variant. Each group was tested in triplicate. Final panel (bottom right) displays total peptide response for all donors.

**Figure 4 pharmaceutics-14-01833-f004:**
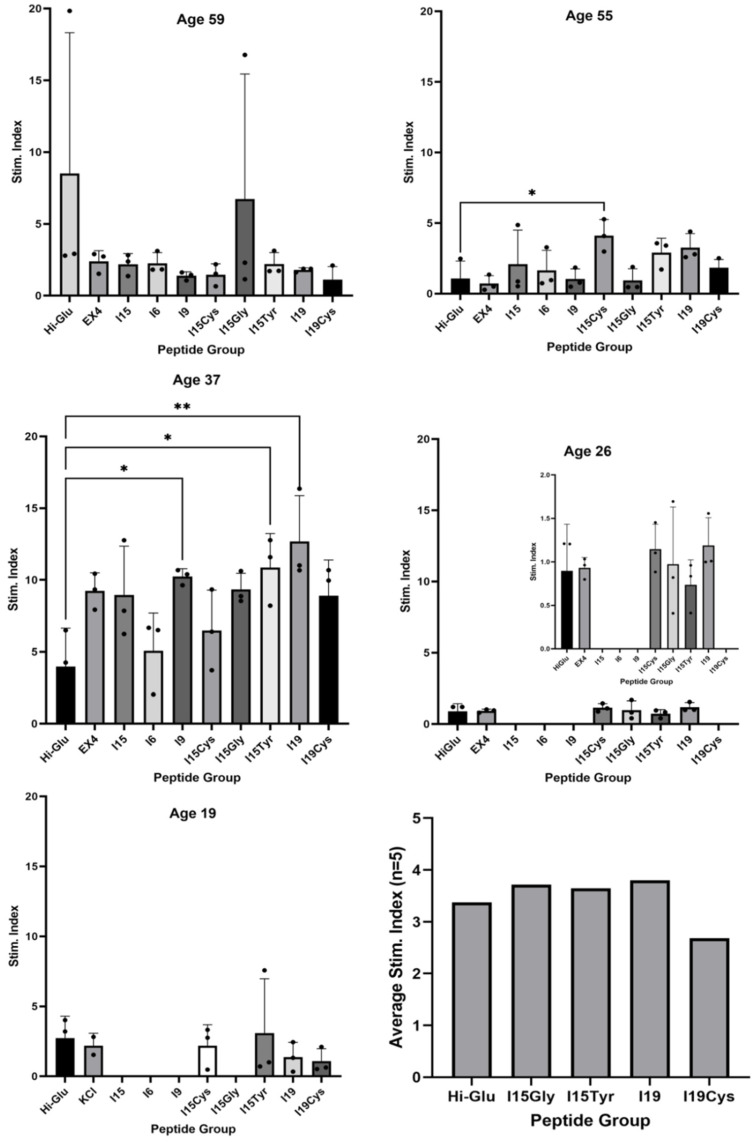
Stimulation index shows fold-increase in insulin from the low-glucose to high-glucose wells (individual samples appearing as black dots over the columns, “•”). One-way ANOVA was used to compare the stimulation index of each group to the negative control, “Hi-Glu” (significant differences between means are shown by “*”, two standard deviations from the mean as “**”, *p* < 0.05). Final panel (bottom right) displays total stimulation index for all donors.

**Figure 5 pharmaceutics-14-01833-f005:**
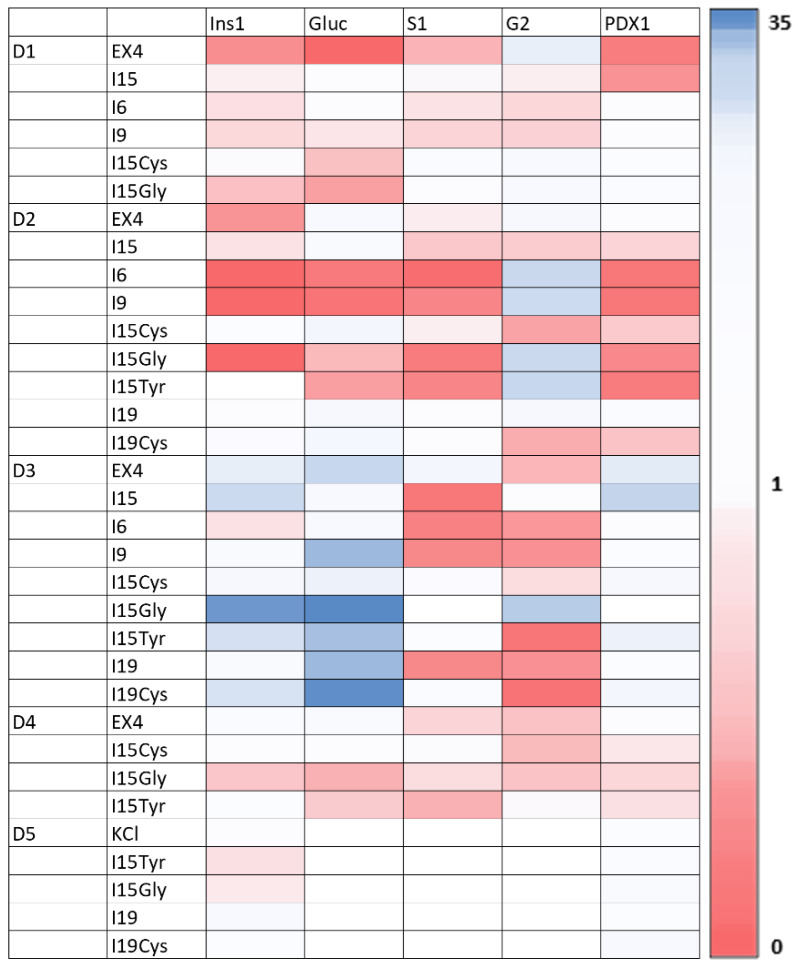
Heatmaps of islet gene expression. Bars are computed using the ddCt method, with the y-axis representing $R=2^{-\Delta \Delta Ct}$. Side legend shows the colour scale, with blue indicating upregulation, and red downregulation.

**Figure 6 pharmaceutics-14-01833-f006:**
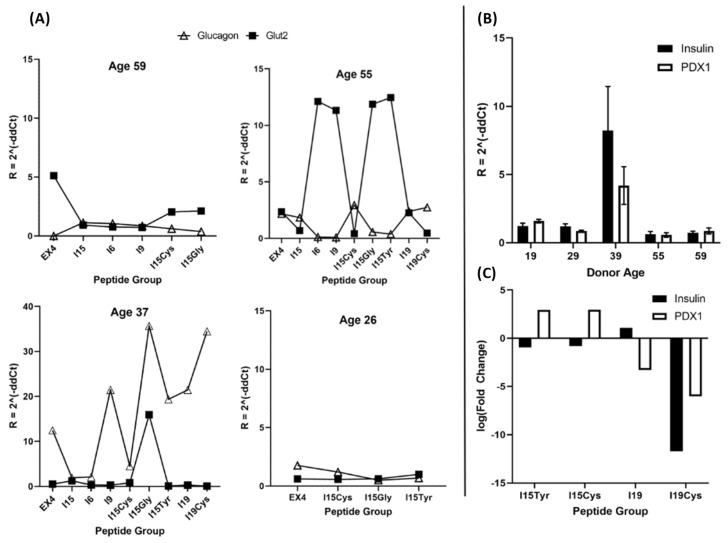
Panel (**A**) Expression of Glucagon (Δ) and Glut2 (■) genes, quantified relative to GAPDH, across peptide groups. These two genes appear to activate in opposition, though donor 3 (age 37) showed simultaneous activation of Glucagon and Glut2, along with producing the most insulin overall in ELISA tests. (**B**) Gene expression quantified relative to GAPDH. (**C**) Fold change for insulin and PDX1 expression at 1 μg/mL vs. 100 μg/mL peptide concentration. Dose-response gene expression tested on donor 5 (age 19).

**Table 1 pharmaceutics-14-01833-t001:** Human islet donor profiles. All subjects were non-diabetic and COVID-19 negative.

Donor	Islet Donor Information									
	Age	Islet Size Index	Purity (%)	Islet Viability (%)	HbA1c (%)	BMI	Cause of Death	Height (in)	Weight (lbs)	Gender
1	59	0.93	90–95	95	5.0	24.5	Stroke	68	161	Female
2	55	0.79	90–95	95	5.6	28.4	Stroke	68	187	Male
3	37	1.25	90	95	5.8	23.9	Head Trauma	76	198	Male
4	26	1.18	95	95	5.9	30.3	Head Trauma	70	209	Male
5	19	1.17	90	95	5.8	23.1	Head Trauma	71	157	Male

**Table 2 pharmaceutics-14-01833-t002:** mRNA primer sequences used for qPCR investigation of gene regulation related to the insulin secretion pathway of human pancreatic islets.

Gene	Function	Forward Primer Sequence	Reverse Primer Sequence
Insulin	Encodes preproinsulin	GAA-CGA-GGC-TTC-TTC-TAC-AC	ACA-ATG-CCA-CGC-TTC-TG
Glucagon	Encodes preproglucagon	ACC-AGA-AGA-CAG-CAG-AAA-TG	GAA-TGT-GCC-CTG-TGA-ATG
SUR1	Membrane protein; target of antidiabetic drugs	CGA-TGC-CAT-CAT CAC-AGA-AG	CTG-AGC-AGC-TTC-TCT-GGC-TT
GLUT2	Transmembrane carrier protein	CTC-TCC-TTG-CTC-CTC-CTC-CT	TTG-GGA-GTC-CTG-TCA-ATT-CC
PDX1	Insulin promoter factor 1	ATG-GAT-GAA-GTC-TAC-CAA-AGC	CGT-GAG-ATG-TAC-TTG-TTG-AAT-AG
GAPDH	Catalyzes glycolysis, can activate transcription	CAC-CCA-CTC-CTC-CAC-CTT-TG	CCA-CCA-CCC-TGT-TGC-TGT-AG

**Table 3 pharmaceutics-14-01833-t003:** Custom designed and synthesized INGAP-P variants for testing on islet insulin secretion with shorthand ID.

Peptide ID	Variant	Sequence	Interest
I15	INGAP-P	N’-Ile-Gly-Leu-His-Asp-Pro-Ser-His-Gly-Thr-Leu-Pro-Asn-Gly-Ser-C’	INGAP’s bioactive region (already proven)
I6	INGAP-P conserved motif	N’-Ile-Gly-Leu-His-Asp-Pro-C	Synergistic effect with I9 conserved motif
I9	INGAP-P specific motif	N’-Ser-His-Gly-Thr-Leu-Pro-Asn-Gly-Ser-C’	Synergistic effect with I6 specific motif
I15Cys	Cyclic INGAP-P	N’-Ile-Gly-Leu-His-Asp-Pro-Ser-His-Gly-Thr-Leu-Pro-Asn-Gly-Ser-Cys-C’	Efficiency of the cyclization method (with cysteines)
I15Gly	Modified INGAP-P (C-terminal)	N’-Ile-Gly-Leu-His-Asp-Pro-Ser-His-Gly-Thr-Leu-Pro-Asn-Gly-Ser-Gly-C’	Effect of glycine at the C-terminal amino acid on the ligand/receptor mechanism
I15Tyr	Modified INGAP-P (C-terminal)	N’-Ile-Gly-Leu-His-Asp-Pro-Ser-His-Gly-Thr-Leu-Pro-Asn-Gly-Ser-Tyr-C’	Effect of tyrosine at the C-terminal amino acid on the ligand/receptor mechanism
I19	Modified INGAP-P	N’-Cys-Cys-Ile_Gly-Leu-His-Asp-Pro-Ser-His-Gly-Thr-Leu-Pro-Asn-Gly-Ser—Cys-Cys-C’	Effect of making a longer peptide with hydrophobic amino acids
I19Cys	Cyclic modified INGAP-P, longer peptide	N’-Cys-Cys-Ile-Gly-Leu-His-Asp-Pro-Ser-His-Gly-Thr-Leu-Pro-Asn-Gly-Ser—Cys-Cys-C’	Efficiency of the cyclization method, combination of longer peptide (19) and cyclization

## Data Availability

The data presented in this study are available upon request from the corresponding author.

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
