# Peer review of "INGAP-Peptide Variants as a Novel Therapy for Type 1 Diabetes: Effect on Human Islet Insulin Secretion and Gene Expression"

_pharmaceutics, 2022, doi:10.3390/pharmaceutics14091833_

Round 1

Reviewer 1 Report

Dear Authors,

 Congratulations on this well-made paper. Your research has the potential to improve the range of possible therapies for Type 1 diabetes. Scientific design is linear and consistent, with well-written explanations.

 You described in great detail the pancreatic islets isolated from human donors, which provide a more useful model for gene expression studies than is typically possible with conventional insulin-secreting cell lines (i.e., INS-1 or RIN cells of nonhuman origin).

The writing of results is well-organized and easy to follow, making it a pleasure to read. The tables and figures are complete, and the study’s limitations have been appropriately acknowledged. In my opinion, the manuscript can be accepted for publication as it is.

Best regards

Author Response

The authors kindly thank Reviewer 1 for their time and consideration in reading this article; we appreciate your response and recommendation for publication.

Reviewer 2 Report

The study was well designed and carried out, the authors present some interesting results that indicate of INGAP-P can elevate insulin secretion, and identifies several variants for further investigation, based on their ability to impact islet signaling genes. The sample size is not large due to the difficulty in obtaining human islets and the authors recognise this and the likelihood that this may affect the conclusions of the study.

Author Response

We are grateful for Reviewer 2’s comment and understanding of the challenges involved. We hope this work serves as a steppingstone to in-depth investigations of the candidates identified.

Reviewer 3 Report

In this manuscript, Porter et al. and colleagues performed a custom synthesis of INGAP-P variants and screened for their potential to improve the beta cell function and viability. They identified four peptides that show a potential to stimulate insulin secretion and genes that are involved in the insulin secretion pathway.

Major points

1.    In this reviewer's opinion, the main limitation of this study is the lack of in-depth functional analyses using in-vitro studies and in vivo transplantation models.

2.    It is well established in the field that INGAP-P has the potential to improve insulin secretion and cell proliferation. Did the authors test the long-term effect of INGAP-P on human islets?

3.    In line with the data on insulin secretion, it would be more informative to see how these new custom peptides could potentiate the kinetics of insulin secretion by perifusion studies.

4.    Although the authors claim the difference in insulin stimulation they observe between different donors is because of heterogenicity, if this is the case, maybe the authors could perform some experiments using a human beta cell line (eg, Endoc BH1).

5.    It is a well-known fact that INGAP-P can induce cell proliferation, so it would be more informative to see whether the custom peptides show similar properties in activation of cell proliferation.

6.    In order to evaluate the value of these peptides in islet cell transplantation studies, it could be important to show how the treatment of these peptides could preserve islet engraftment and function in vivo models of T1D.

Minor points.

1.    Check for grammar in line 74

Author Response

In this manuscript, Porter et al. and colleagues performed a custom synthesis of INGAP-P variants and screened for their potential to improve the beta cell function and viability. They identified four peptides that show a potential to stimulate insulin secretion and genes that are involved in the insulin secretion pathway.

Major points

  1. In this reviewer's opinion, the main limitation of this study is the lack of in-depth functional analyses using in-vitro studies and in vivo transplantation models.

The authors agree that in vivo transplantation models are critical in the translation of therapies to patients, in fact the INGAP molecule has been studied in in clinical human trials, as discussed in the manuscript [1]. Based off those works, this exploratory study aimed to identify potential agents for the subject of such in depth analyses, as suggested by the reviewer. The number of peptides tested, however, meant there was a large cost and time required for synthesis, GSIS and gene testing. Therefore, we find the study a valuable effort in attempting to identify candidates for deeper study. In accordance with the reviewer’s comment, however, we have revised the beginning of the limitations section to include the following:

“The primary limitation with identifying therapeutic compounds for T1D therapy, as in this study, is the need for extensive in-vitro analysis and in-vivo transplantation models. Due to the steps involved with synthesis and initial screening evaluation, along with the number of compounds investigated, the authors found it necessary and reasonable to first identify agents of interest to qualify for in-depth analysis.

  1. It is well established in the field that INGAP-P has the potential to improve insulin secretion and cell proliferation. Did the authors test the long-term effect of INGAP-P on human islets?

The authors whole heartedly agree that such longer-term studies would inform the ability of INGAP-P variants to promote graft transplantation survival. Cells could be exposed to peptides for 1-5 days and tested for effects on proliferation as well as secretion. This data set would indeed be included in the next study of these peptides effects on immortalized insulin producing cells, however with native islets being terminally differentiated, their rate of beta-cell proliferation is low in comparison to other cell types. Additionally, native human donor islets such as those used here have a limited lifetime of functional secretion following shipping and recovery from transport media, and cell lysis for qPCR precludes such longer-term experiments. We therefore opted to focus the scope of this immediate study on the peptides’ ability to stimulate insulin compared to the known stimulator exendin-4 in a short-term fashion and conduct peptide culture incubation studies with immortal cell lines in a follow up study.

  1. In line with the data on insulin secretion, it would be more informative to see how these new custom peptides could potentiate the kinetics of insulin secretion by perifusion studies.

We fully agree that deeper understanding can be gained from dynamic assays (such as the kinetic binding experiments currently in progress for a follow-up study using surface plasmon resonance). To ensure the reviewer, it is noteworthy to mention that we have already undertaken these investigations which will be the subject of anther manuscript in the near future, as they exceed the scope of current study. So far, our preliminary perifusion tests of human islets using known secretagogues (Fig. 1) displayed very much similar line shapes, despite varying degrees of insulin released. Based off these initial observations it was decided to first focus on quantifying the amount of insulin released during the primary response as a gauge for evaluation. This description is now included at the end of section 3.2 (lines 314-319), and above figure included in supplemental information.

Please also see the figure in the attached file' authors' cover letter'. 

  1. Although the authors claim the difference in insulin stimulation they observe between different donors is because of heterogenicity, if this is the case, maybe the authors could perform some experiments using a human beta cell line (eg, Endoc BH1).

      As briefly mentioned in the main text; we do believe that using such an immortal cell line should be the next step in complete exploration of the effects of these compounds, prior to any in-vivo transplantation work (lines 425-427):

“Other non-β islet cells would contribute mRNA following lysis, leading to uncertainty of the regulation of insulin secretion related genes. This could be addressed by using immortalized cell lines or separating islet cells to specifically interrogate β-cells.”

However, to clarify to the readers that for the authors one of the major takeaways from this study was that immortal cell lines should be used for more consistent screening studies going forward, the following was added to the main text in the conclusion section (lines 459-461):

“Future screening of therapeutic compounds could be conducted on immortalized human insulin producing cell lines (such as EndoC-H1) in order to reduce variability.”

Such human insulinoma cells, however, can be difficult to obtain, require optimization of culture and spheroid formation, and do not ultimately share the cytoarchitecture or gene expression of human islets.

Nevertheless, in line with this suggestion, the authors have begun testing some of the identified compounds on human beta cell line EndoC-H5 spheroids (Fig. 2: just for the reviewer). While initial GSIS testing has shown reduced variation of insulin secretion in comparison to the islets from individual human donors in this study, we have yet to identify significant improvements based on the dosages and durations of peptide exposures used. It is for these and above reasons we feel the data lie adjacent to the specific scope of the current article and will fit better in the complete study of variant peptides’ effect on these established cell lines.

Please also see the figure in the attached file' authors' cover letter'. 

  1. It is a well-known fact that INGAP-P can induce cell proliferation, so it would be more informative to see whether the custom peptides show similar properties in activation of cell proliferation.

We fully agree this should be the subject of further studies, however focused on short-term insulin secretion for those reasons detailed in the response to comment #2.

  1. In order to evaluate the value of these peptides in islet cell transplantation studies, it could be important to show how the treatment of these peptides could preserve islet engraftment and function in vivo models of T1D.

The authors certainly agree that in-vivo studies will be required to properly evaluate promising T1D treatment candidates. We investigate these compounds’ ability to improve insulin secretion, such that during or following transplantation they could be administered to enhance the impact of grafted islets. Future studies could explore encapsulating islets within scaffolds containing the peptides, and other novel delivery strategies. Of course, if any of these variants additionally show improvements of islet viability with longer-term culture they could also be used for in-vitro pre-treatment of islets or other insulin producing cells prior to implant.

Minor points.

  1. Check for grammar in line 74

Thank you for this comment. We replaced:

“INGAP-P’s mechanism of action is not well understood, nor how interacts with the beta cell" by: “The mechanism of INGAP-P's interaction with the beta cell is not completely understood".

Reference

[1] Rosenburg, L. “Modified INGAP Peptides for Treating Diabetes”, US Patent US20160002310 A1, 2016

Round 2

Reviewer 3 Report

Dear Editor,

This study, "INGAP-Peptide Variants as a Novel Therapy for Type 1 Diabetes: Effects on Human Islet Insulin Secretion and Gene Expression” investigates the therapeutic potential of INGAP-Peptide Variants in improving the function of pancreatic islet cells.  The authors have improved the current version of the manuscript and presented some preliminary supporting data in their response to the reviewer document, which further implicates their conclusions. Finally, I don't have further comments to add to this manuscript.

Best regards,

Farooq